# viromeBrowser: A Shiny App for Browsing Virome Sequencing Analysis Results

**DOI:** 10.3390/v13030437

**Published:** 2021-03-09

**Authors:** David F. Nieuwenhuijse, Bas B. Oude Munnink, Marion P. G. Koopmans

**Affiliations:** Viroscience Department, Erasmus Medical Center, Doctor Molewaterplein 40, 3015 GD Rotterdam, The Netherlands; d.nieuwenhuijse@erasmusmc.nl (D.F.N.); b.oudemunnink@erasmusmc.nl (B.B.O.M.)

**Keywords:** metagenomics, virome, bioinformatics, NGS analysis, shiny, visualization, data analysis

## Abstract

Experiments in which complex virome sequencing data is generated remain difficult to explore and unpack for scientists without a background in data science. The processing of raw sequencing data by high throughput sequencing workflows usually results in contigs in FASTA format coupled to an annotation file linking the contigs to a reference sequence or taxonomic identifier. The next step is to compare the virome of different samples based on the metadata of the experimental setup and extract sequences of interest that can be used in subsequent analyses. The viromeBrowser is an application written in the opensource R shiny framework that was developed in collaboration with end-users and is focused on three common data analysis steps. First, the application allows interactive filtering of annotations by default or custom quality thresholds. Next, multiple samples can be visualized to facilitate comparison of contig annotations based on sample specific metadata values. Last, the application makes it easy for users to extract sequences of interest in FASTA format. With the interactive features in the viromeBrowser we aim to enable scientists without a data science background to compare and extract annotation data and sequences from virome sequencing analysis results.

## 1. Introduction

The use of metagenomic sequencing to explore the virome of complex samples such as stool, sewage, and environmental samples has increased over the years [1]. Alongside the increase in virome studies, a plethora of viral metagenomics processing workflows have been created [2]. The results of experiments in which complex virome sequencing data are generated are difficult to visualize and unpack for a person without programming experience. After processing of the raw sequencing data by next-generation sequencing (NGS) processing workflows, the usual output consists of contiguous sequences (contigs) in FASTA format and an annotation file linking the contigs to a reference sequence or taxonomic identifier in tabular format [3,4,5,6]. For example, Virusfinder outputs several text files containing annotations and contig sequences as results [3], and VirFind outputs tabular annotation and FASTA files for viral and non-viral annotations [5]. Further unpacking of results and getting an overview of which viruses are found in which samples is especially difficult when these data are spread over multiple tables containing many annotations. Many of the workflows generate a summary file and figures containing a selection of the annotated sequences from the sample based on predefined selection criteria. For example, SURPI generates a summary annotation file specifying read counts and contig coverage but relies on Excel for annotation summarizing and comparison [4]. Interestingly, virus identification pipeline (VIP) produces an html report that can be browsed interactively for individual samples and viruses but does not allow for comparison of samples [6]. A first and routine step in virome analysis is an annotation quality check. If quality thresholds are set upfront in the sequence annotation workflow, depending on the virus and the intentions of the user, these settings can be too stringent or too relaxed, resulting in either false negatives or false positives [7,8,9]. Therefore, manual inspection and manual filtering using custom quality criteria are often needed, which means that the workflow has to be rerun with different settings or, if possible, the unfiltered annotation results can be filtered using custom scripts and further analyzed using visualization tools.

Several visualization tools have been made to view annotation results while staying relatively agnostic of the metagenomic sequencing workflows such as MEGAN, Pavian, Krona, PanViz, MetaViz, and Anvi’o [10,11,12,13,14,15]. MEGAN and Pavian perform very extensive analyses, but only accept specific input formats, making them less flexible to use with different kinds of analysis workflows. PanViz, MetaViz, and Anvi’o are tailored to the analysis of bacterial metagenomic data and are less well suited for virus data. Krona is very flexible and easy to use but cannot be used to easily compare multiple samples side by side. Paid software such as Geneious (https://www.geneious.com (accessed on 8 March 2021).) and CLC bio (https://digitalinsights.qiagen.com (accessed on 8 March 2021)) are also available but require expensive licensing and cannot be customized or cannot visualize BLAST outputs in a concise manner.

In collaboration with end-users, we developed viromeBrowser, a virome browsing app that works through the steps they commonly use when interpreting sequencing outputs. The browser imports multiple annotation files and a corresponding FASTA files containing annotated contigs, addressing three common processes: (1) annotation quality assessment, (2) dataset visualization and interpretation, and (3) extraction of sequences with a specific annotation. An iterative design process was employed with end-users to allow intuitive browsing, selecting, and exporting of specific sequences and selections, as well as visualization of these sequences combined with metadata. 

## 2. Materials and Methods

The virome browser was written in the R programming language and makes use of the Shiny [16] web application R package. The interactive visualizations are created using the rbokeh package [17]. Rsamtools [18] is used to handle the FASTA files, and open reading frame prediction is performed by splitting the contig sequence based on the presence of a stop condon using the Biostrings package [19]. The application was packaged for easy installation following the guidelines of the O’Reilly R Packages book [20] and is available on the R CRAN platform. Examples of virome datasets with metadata were used to present the tool to end-users and invite feedback for further optimization. In total, we went through several iterations before finalization of the application.

## 3. Results

The viromeBrowser is implemented in Shiny [16] a web application framework and depends on several other R packages as shown in the package description at https://CRAN.R-project.org/package=viromeBrowser (accessed on 8 March 2021). Even though Shiny can be used to create web applications, the server and client part of the application can also be set up locally for analysis of data in situations where sharing may not be not allowed, such as in clinical settings. This setup also allows an institute to run the computational heavy part on a centralized powerful computer while running the lightweight client on the user’s computer. The user interface (UI) elements of the viromeBrowser have been made modular to allow for easier expansion of the app, using the Shiny functionalities for module development. A video demonstrating the complete application is provided in Appendix A.

### 3.1. Data Input 

The app is separated into three main parts, which are listed as separate menu items. The first part allows data to be loaded into the app by selecting a FASTA file containing contig sequences, a contig annotation file, a binary alignment (BAM) file containing mapped reads to contigs, and a metadata file (Figure 1). These files need to be in the format specified in the packages vignette. Briefly, the contigs have to be in regular FASTA format with the FASTA headers exactly matching the contig identifiers in the annotation and read mapping files, the contig annotation files need to be in BLAST tab-separated format, the read mapping files need to contain the mapped reads to the aforementioned contigs, and the metadata file needs to be in tab-separated format in which the file names are in the first column and the sample characteristics are listed in each additional column. Currently, only default tabular BLAST-like output is supported, but other tabular formats can be implemented by creating a novel import function. To determine the taxonomic lineage of each annotation based on the associated taxonomic identifier, the application uses the “rankedlineage.dmp” file, which is downloaded automatically from the NCBI taxdump database at ftp.ncbi.nlm.nih.gov/pub/taxonomy/new_taxdump/ (accessed on 8 March 2021). For contigs with multiple associated annotations, which is not unusual for BLAST output, a lowest common ancestor (LCA) taxonomic lineage is determined by finding the lowest order taxon that is present in all annotations. If no LCA can be found, the contig is annotated as “root”. The LCA calculation is performed after annotation quality filtering to avoid spurious hits from interfering in the process.

### 3.2. Interactive Quality Assesment 

The second part of the app displays an interactive heatmap based on the annotations provided in the contig annotation and metadata files. The initial data of the heatmap is based on default quality settings aimed at highlighting contigs larger than 500 nucleotides with more than 90 percent identity over a length of 500 nucleotides or more to the chosen references. Workflows may differ with regards to annotation quality parameters, and therefore the default quality settings are part of the data import function and can be customized accordingly. Users can override the default quality thresholds by filling in other values in the quality threshold tab. This gives full control over the filtering of the data, allowing users to browse contigs for virus discovery or set the filters such that only results with high confidence are visible. Annotation settings could, for instance, be set to specifically target low sequence identity annotations to filter out novel viruses, or to high-sequence identity to continue with high-certainty annotations only. On the other hand, annotation settings can be made stricter for diagnostic purposes to prevent false positives.

Further selection of specific annotations of interest can be done by using the rectangular selection tool in the interactive heatmap. The selection made in the heatmap will propagate to the rest of the app, enabling the user to zoom in to a specific annotation, sample, or sample characteristic of interest. Deselecting all tiles in the heatmap will reset the selection, resulting in the selection of all annotations in the current view.

### 3.3. Metadata-Guided Sample Comparison

Once the quality settings have been defined, the next step is the interpretation of the obtained results in combination with the metadata of the prepared experiment or sample cohort. This part was implemented in the viromeBrowser by an interactive heatmap, which can be used to stratify, filter, and group samples based on the provided metadata file. The interactive heatmap can also be used to compare multiple sample annotation files in a single overview and from different points of view. The interactivity is enabled by three dropdown menus (Figure 2). In the first menu, the factor can be chosen by which the samples in the heatmap will be stratified. In the second two menus, the value by which the heatmap tiles are colored and the taxonomic level by which the heatmap has to be drawn can be selected. The fill options are either by number of contigs, absolute number of reads, or relative number of reads, scaled by the total number of reads in the BAM read mapping file.

### 3.4. Sequence Extraction and Downloading

Another functionality of the app is further inspection and downloading of specific contigs sequences. A table shows the annotations based on the selections made in the interactive heatmap and quality thresholds in the previous tab. Sequences can be selected from the table for further inspection or saving, but it is also possible to save all sequences from the table by selecting the download all filtered button.

Users can continue to zoom in on a single contig in the app by selecting one or more contigs from the table and continuing to the contig information tab (Figure 3). In this tab, a single contig is displayed for which open reading frames (ORF) are predicted by performing a canonical stop codon lookup and using these to split each frame into ORF fragments. The ORFs are represented by arcs on the contig and are predicted for all six frames, and small spurious ORFs can be filtered by setting a minimal ORF size. The visualization allows users to perform a quick check of the expected ORF structure of an annotated virus. Individual ORFs can be viewed under the ORF information tab, and nucleotide or amino acid sequences can be directly selected and copied. Alternatively, all displayed ORFs can be collected in the ORF collection table and saved in FASTA format. For further analysis, it is sometimes useful to only extract certain ORFs from a genome, which is possible by separately saving ORFs in FASTA format.

## 4. Discussion

The viromeBrowser can be used to interrogate the analysis results of complex metagenomic sequencing experiments such as viromes of stool, wastewater, or environmental samples. A unique feature is that the viromeBrowser starts with filtering of analysis results by quality, allowing less or more stringent selection of annotations, which is important given the large diversity differences of viruses and the different interests of users, be it virus discovery or viral diagnostics, bridging the gap between research settings and diagnostic usage. The former requires a broader view in which quality parameters can be more lenient, which is useful for virus discovery, while the latter requires stringent quality threshold to prevent false positive results. This limits the use of viromeBrowser to workflows that output unfiltered annotation results of contigs as part of their analysis, which makes it incompatible with read-based annotation workflows such as SURPI [4]. 

A feature that was added based on end-user input was the possibility to compare viromes based on added metadata parameters, for instance, in a setting where the goal is to find differences in the virome of patient and control groups, linking annotations with the associated contig sequences and custom extraction of selected sequences for further analysis. This allows interactive visualizations to aid the user in making a manual selection of the data that can be used in further analyses such as primer design, phylogenetic analyses, or variant analysis. An alternative tool with strong visualization and result selection features is Genome Detective [21], which provides user-friendly NGS data processing and visualization via a web-interface. However, for settings where data cannot be shared externally, local installation of Genome Detective can only be done by paying a license fee and paying a fee per sample analyzed.

Future improvements could be made on the implementation of other annotation formats to create a more flexible interface for annotation data input. Additional improvements could be added to obtain more visualization options for detailed contig analysis, such as contig coverage plots and variant visualization by importing read alignment files.

To keep the application interactive and lightweight, the pre-required raw data processing and annotation steps are not performed by the viromeBrowser, which is a disadvantage for users without a running NGS processing workflow. To address the need for data processing by users without their own raw data processing capacity, the European Bioinformatics Institute has developed datahubs in which raw data can be uploaded and analyzed with several standardized workflows [22]. After the preprocessing and annotation steps have been performed, the viromeBrowser can be used to inspect the results.

## 5. Conclusions

In conclusion, here we present viromeBrowser, an interactive application to browse through the annotation results of viral metagenomic sequencing experiments. Interactively selecting viral annotations of choice and manually tuning quality thresholds should make it easier for scientists with little programming experience to analyze complex metagenomics data. Facilitating separating and downloading of contigs of interest will make it easier to perform follow-up analyses with these sequences. The viromeBrowser is implemented as an R package, which is distributed by the R CRAN platform. Future updates will be available from the same platform.

## Figures and Tables

**Figure 1 viruses-13-00437-f001:**
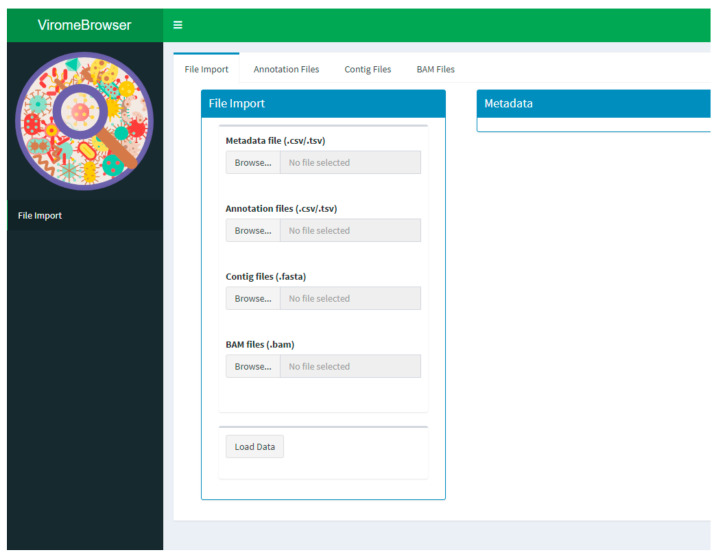
Example screenshot of the file import screen. Metadata, annotation files, and contig files are uploaded and processed in the file import screen. An excerpt of the metadata, the annotation files, and the contig files can be visualized under the corresponding tabs.

**Figure 2 viruses-13-00437-f002:**
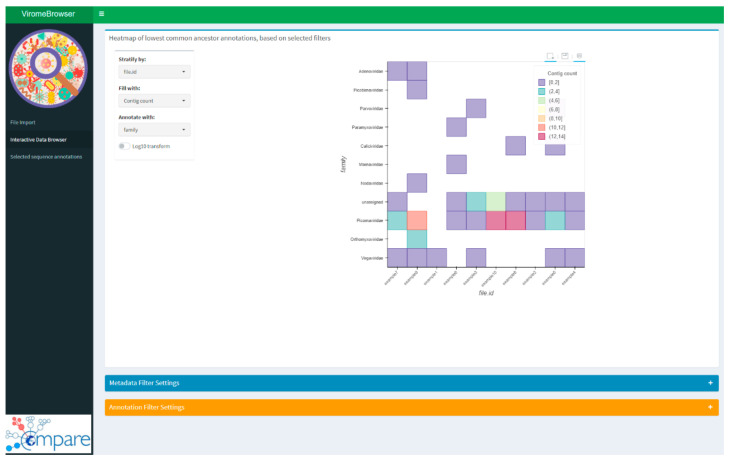
Screenshot of the interactive data browsing heatmap. Stratification variable and filter criteria can be selected in the browser settings window. Annotation quality filter settings are available as a drop-down menu in the bottom of the page. Hovering over the heatmap shows the contig annotations and the number of contigs or read counts. Selecting tiles of the heatmap results in selection of only those contigs for further analysis and downloading.

**Figure 3 viruses-13-00437-f003:**
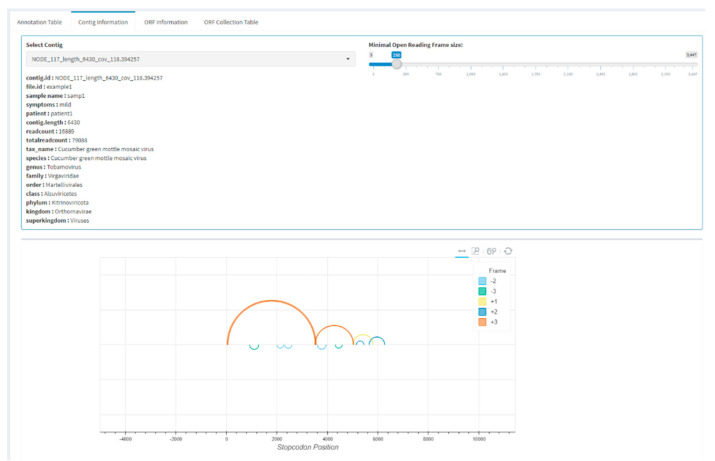
Example screenshot of the sequence information tab. Contigs can be selected from a table of previously selected annotations. Further inspection can be by visualization of the open reading frame (ORF) structure and downloading of individual ORFs or the complete contig.

## Data Availability

Not applicable.

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
