# Peer review of "viromeBrowser: A Shiny App for Browsing Virome Sequencing Analysis Results"

_viruses, 2021, doi:10.3390/v13030437_

Round 1

Reviewer 1 Report

Nieuwenhuijse et al present a potentially useful tool, a lightweight browser of ORFs in contigs. The introduction to the paper gives a convincing rationale for writing this tool, and thoroughly advertises the use of Shiny. The tool is indeed very easy to handle, and may potentially help understand the virome.

I have two major points: one is conceptual (what should be displayed and how), the other technical (the tool does not seem to work properly - or at least behaves contra-intuitively).

POINT 1: I think the tool might be even more useful if the authors consider user needs in a broader context. The program, as it is, leaves uncovered rather many common tasks in virome research. The authors entirely disregard the existence of frequency tables, and the possibility of simultaneous processing of several samples of common origin (a set of samples compared within a study, be it a set of individuals, or sites). This may be for instance an association study of the virome with a condition that need not be viral (sputum virome with exacerbation of COPD, gut virome and Crohn disease etc). There a very complex viromes must be explored in dozens or even hundreds of samples. Then the frequency (OTU-like) tables are produced, containing the counts of reads mapped to individual contigs. A common task for the researcher is then to browse through "differentially expressed" contigs or their clusters and assess whether the significant difference in read coverage between cases and controls belongs to anything interesting. Good inspiration might be the Shiny Phyloseq app for bacteriomes.

To this end, the tool would greatly benefit from adding the functionality for comparing data across samples, i.e. from importing and processing read abundance tables. Now it is entirely impossible to discern an important signal of a causative virus from ever-present background caused for instance by the "kitome".

POINT 2: I might not have understood the program well - but when testing it, I detected some very strange behaviour.  I evaluated the app in parallel with manual inspection of the source files on Github.

  1. a) The test file "example 1" has 8 contigs. However, the tab "Annotation Files" shows only Contig1 for each file.id (it says "The first line of each imported and parsed file is shown"). Why? Where are the remaining contigs and what is the reason for hiding them? If you worry about space, pack them and make a "+" icon for unpacking.
  2. b) Interactive data browser: I spent nearly half an hour figuring out how to switch off the filter. First you must go down and unclick the "Use default settings" (that however does NOT display the default setting! It displays "All" in all fields, which is not true) and then go back to the top and "Apply Filter", although there is actually NO FILTER SPECIFIED. This use of the "Apply filters" is extremely contra-intuitive and user-unfriendly. It is incorrect to actually apply criteria buried deep in the paper while displaying "all" in the filter characteristics. This must be fixed.
  3. c) Sequence Information tab: Again, I was puzzled by the fact that the contigs are first filtered in the seemingly unrelated tab "Interactive Data Browser" and only then displayed using a tab "Sequence Information". It should be a visualization tool, so please make sure the users understand what they visualize and what the sequence of events is before they browse the result. The left pane is nearly empty, kindly use the empty space for listing the parameters of filtering and possibly also the results of filtering.

 Minor points:

- "Currently, BLAST-like tabular format is allowed" (vignette) - this is a great disadvantage. Blast produces many other, better outputs than tables (xml, for instance); parsing one type of annotation into another is nothing that can be done by users inexperienced in bioinformatics.

- Why the Sequence Information -> Annotation Table cannot be expanded vertically? Only two rows are visible. This is extremely annoying.

- Export All tool exports only displayed contigs, definitely not all.

- The graphics of the heatmap should be more condensed. Too large rectangles add no information, and may easily fill even a big screen.

- What are the colors in the heatmap? Why there is no legend? The use of heatmap suggests some quantitative information, when there is actually none. An abundant virus might realistically assemble into one full contig (parvovirus) and this single contig will score less than countless fragments of a poorly covered virus that will score one point for each contig. The abundance info should be definitely incorporated.

In conclusion, this piece of software somehow lacks thorough documentation (so this reviewer - despite being an author to another virome analysis tool - did not understand what to do), and it may be significantly improved by adding quantitative aspects of virus mapping, similar to its counterparts from bacteriology (Shiny Phyloseq). In my opinion, the paper itself it written well, but the piece of software would deserve some additional work before the paper can be published. The idea behind the work is commendable, and the authors are indisputably one of the best virology teams in Europe. I am sure they can easily improve the tool in order to help the community!

Reviewer 2 Report

Line 4 - Must be full address

Line 6 – Delete …(200 words)

Line 9 - next generation sequencing (NGS) … I propose to replace it with «high throughput sequencing»

Line 12, 13 - Wrong indentation between lines.

Line 18, 19 - Wrong indentation between lines.

Line 22 - I suggest adding the word «Bioinformatics» and change NGS to HTS (high throughput sequencing).

Line 25 to 127 - Align text to width

Line 48, 49 - Wrong indentation between lines.

Line 201 – Software … The first letter is in bold.

Line 211 – Add DOI

Line 214 - The words in the title of the article begin with lowercase and uppercase letters.

Line 230 - The words in the title of the article begin with lowercase and uppercase letters.

Line 238 - The words in the title of the article begin with lowercase and uppercase letters.

Line 255 - The words in the title of the article begin with lowercase and uppercase letters.

Currently, there are many programs for processing viral metagenomes. But most of them are aimed at confident users with high knowledge of bioinformatics. In this article, the authors offer a tool for entry-level users, although advanced users may find this tool useful. In a literature review, the authors demonstrate the available programs for processing viral metagenomes and point out the uniqueness of their development. The authors clearly describe the functionality of the program, although there are oversights, which are described below.

Since the package is aimed at the initial level of knowledge in bioinformatics, I think it will be difficult to understand where to get the Annotation files. It is necessary to clearly make it clear to novice users about preparing the necessary files. I suggest adding an example sample metadata and annotation files to the Supplementary as well. To check the browser, R latest version 4.0.3 and R-Studio Version 1.3.1093 were installed, immediately there were problems with packages such as: Biostrings and Rsamtools, they are easily installed from Bioconductor, but the authors have no information about this. Perhaps the information will appear in the technical section, but I would like this moment to be covered in the manual. Only from the video is it clear on which version of R the browser was loaded.

All input sequences must be annotated, which cannot be done for all viral sequences. Hence the complexity of loading data. For example, after the SPAdes assembler, some of the contigs are not annotated in BLAST and you will have to sort according to the annotation file.

In the File Import - Contig files tab, it would be convenient to place a column with the number of contigs in the file.

In the Sequence information - Annotation Table tab, it is impossible to expand the annotation window, which brings inconvenience.

In the ORF Information tab, why is there a choice of changing the ORF length? It is not clear to me whether ORF exists and is limited to stop codons or not. It would be much more convenient to view the entire contig and all the ORFs, as it is implemented in EasyFig.

In the future, I would like to wish the authors to provide high-quality support for their product. it often happens that the developed programs cease to be updated and supported, losing their relevance.

Round 2

Reviewer 1 Report

I am afraid the revised app does not work.

Can I ask the authors to borrow someone's laptop, preferrably never used for any bioinformatics, and try?

1. The vignette never loaded

> vignette("viromeBrowser") Warning message: vignette ‘viromeBrowser’ not found

2. I was not able to get past the data upload. I used the supplementary dataset, duly waited for upload. The files were there but no new menu items appeared.
The vignette on Github promises "Two new menu items will also appear in the sidebar on the left, click on the "Interactive Data Browser" tab to continue.". Actually, it was not the case.

My R version is 3.6.3 (2020-02-29) on a Windows 10 Pro workstation with abundance of RAM and processor cores. That should not be a hardware issue. And now I am a bit reluctant to try on my linux R server...

Kindly check and retest the software; I am looking forward to having the opportunity to re-review the manuscript!

Author Response

Dear reviewer,

I'm sorry for the inconvenience.

  1. I realized that when installing from GitHub the vignettes are not generated automatically, unlike installing from CRAN. Running install_github("dnieuw/viromebrowser", build_vignette = TRUE, force = TRUE) should fix that and force a reinstall. I have updated the GitHub main page to indicate that.
  2. I have asked a colleague to install the app from GitHub and run it with the same files from the zip file I have attached as a supplemental file. She noticed that the rankedlineage.dmp file was not included.

Did you download this file from ftp://ftp.ncbi.nlm.nih.gov/pub/taxonomy/new_taxdump/new_taxdump.zip?

(I didn’t include it because it is rather big and freely available online).

After downloading that zip and extracting the rankedlineage.dmp file from it and importing the CSV, bam, fasta, and metadata file she was able to get the two additional menus appearing and the app working.

The hardware you are using should not be a problem indeed, I have a regular desktop Windows 10 pc that can run the app and my colleague uses a MacBook.

I hope that this fixes the problems running the app. I am looking forward to your further review.

Kind regards,

David Nieuwenhuijse

Round 3

Reviewer 1 Report

Although many issues have been fixed, the app still does not work as expected, kindly see the attached pdf file with screenshots.
